# Microstructural Aspects of the Fabrication of Al/Al_2_O_3_ Composite by Friction Stir Processing

**DOI:** 10.3390/ma16072898

**Published:** 2023-04-05

**Authors:** Sergey S. Malopheyev, Ivan S. Zuiko, Sergey Yu. Mironov, Rustam O. Kaibyshev

**Affiliations:** Laboratory of Mechanical Properties of Nanoscale Materials and Superalloys, Belgorod National Research University, Pobeda 85, Belgorod 308015, Russia; malofeev@bsu.edu.ru (S.S.M.); zuiko_ivan@bsu.edu.ru (I.S.Z.); rustam_kaibyshev@bsu.edu.ru (R.O.K.)

**Keywords:** aluminum matrix composite, alumina nanoparticles, friction stir processing, microstructure, texture, electron backscatter diffraction (EBSD)

## Abstract

The purpose of this work was the examination of microstructural evolution during the fabrication of an Al/Al_2_O_3_ composite by friction stir processing (FSP). In order to obtain new insight into this process, a longitudinal section of the produced composite was studied, and advanced characterization techniques (including electron backscatter diffraction and microhardness mapping) were applied. It was found that the reinforcing particles rapidly rearranged into the “onion-ring” structure, which was very stable against the subsequent dispersion. Specifically, the remnants of the comparatively coarse-scale particle agglomerations have survived even after 12 FSP passes. Therefore, it was concluded that three or four FSP passes, which are often applied in practice, are not sufficient to provide a homogeneous dispersion of the reinforcing particles. It was also revealed that the gradual distribution of the nanoscale Al_2_O_3_ particles throughout the aluminum matrix promoted a subtle reduction in both the portion of high-angle boundaries and the average grain size. These observations were attributed to the particle pinning of grain-boundary migration and dislocation slip.

## 1. Introduction

Aluminum matrix composites offer an attractive synergy of properties, including low density, good strength, and superior thermal stability [1]. Accordingly, these materials are used for high-performance applications in the transportation sector (particularly in the automotive industry), specifically for the manufacturing of pistons, brake rotors, connecting rods, etc. [2]. An important advantage of aluminum matrix composites if compared with other lightweight composites (such as magnesium or titanium-based) is a reasonably low price–performance ratio [3].

Friction stir processing (FSP) is sometimes considered a promising technique for the fabrication of aluminum matrix composites [4,5,6,7]. Due to the solid-state nature of this technology, FSP minimizes interfacial reactions between the reinforcing elements and the aluminum matrix and thus provides good interfacial bonding between these two constituents.

The extensive research in this area has conclusively demonstrated the high efficiency of FSP for the fabrication of aluminum matrix composites [4,5,6,7]. Specifically, a wide range of composites was successfully produced using a variety of both reinforcing particles and matrix aluminum alloys. The reinforcing elements typically comprise SiC [8,9,10,11,12,13,14,15] or Al_2_O_3_ [11,16,17,18,19,20,21,22,23,24,25] phases but may also involve other particles [26,27,28,29,30,31,32,33]. On the other hand, the matrix materials included the 1xxx [8,9,16,29], 2xxx [20,21,22], 5xxx [11,13,27,30,32,33], 6xxx [12,17,23,31], and 7xxx [10,26] series of aluminum alloys.

Moreover, it was found that the agglomeration of the reinforcing particles during FSP represents an essential problem. Specifically, a single FSP pass is usually not sufficient to produce a homogeneous particle distribution. Typically, three to four passes are applied [11,17,18,19,20,21].

It is important to point out that the distribution of the reinforcing particles throughout the FSP-produced composites is normally examined in the *transverse* cross section of the processed material. In this context, it is worth noting that the FSP technology for composite fabrication is based on the preplaced powder strategy. According to this technique, the reinforcing particles (or powder) are distributed throughout the matrix material from the preliminary-drilled grooves, which are preplaced *along* the FSP path. Hence, it would be useful to also check the particle distribution in the *longitudinal* section of the composites. This approach may provide new insight into microstructural evolution during the FSP fabrication of aluminum matrix composites and thus improve our understanding of this process. This was the objective of the present work.

## 2. Materials and Methods

### 2.1. Materials

The commercial aluminum alloy 5182 was employed as a metal matrix material. The material was manufactured by semi-continuous casting using the casting equipment of the Joint Research Center “Technology and Materials” at Belgorod National Research University. The measured chemical composition of the matrix material was Al-4.75Mg-0.3Mn-0.15Zr-0.1Ti-0.1Cr-0.06Si-0.05Fe (wt. %). The cast ingot was homogenized at 360 °C for 24 h, sliced along its longitudinal direction, and then cold rolled to a total thickness reduction of 75%. The final thickness of the rolled sheets was 5.2 mm.

The nanoscale alumina (Al_2_O_3_) powder was used as a reinforcing phase. The powder particles had a nearly spherical shape and an average diameter of 20 nm.

To introduce the Al_2_O_3_ phase into the aluminum matrix, a series of grooves was machined within the aluminum workpieces (Figure 1), in which the alumina powder was filled. The grooves were arranged in a checkboard pattern along the FSP line (Figure 1a) and had a diameter of 1.5 mm and a depth of 1 mm (Figure 1b). The mutual distance between the grooves was 3.5 mm. The approximate volume fraction of the alumina powder $F_{v}$ was evaluated using the equation [9] $F_{v}=V/\left(A\times L\right)$, where V is the total volume of the grooves, A is the cross-section area of the stir zone, and L is the length of the FSP path (Figure 1a). The calculated volume fraction was ≈1%.

### 2.2. FSP Procedure

The workpieces with the preplaced Al_2_O_3_ powder were subjected to FSP using a commercial AccuStir 1004 FSW machine. To maintain consistency with scientific literature, the conventional FSP reference frame was used, which included FSP direction (PD), normal direction (ND), and transverse direction (TD), as shown in Figure 1a.

To minimize mechanical abrasion of the FSP tool during FSP, the tool was manufactured from a tungsten carbide alloy. The geometry and principal dimensions of the tool are shown in Figure 2a.

To produce a fine-grained structure in the aluminum matrix, a comparatively low tool rotation rate of 500 rpm was applied. On the other hand, to enhance the dispersion of Al_2_O_3_ powder, the tool translation rate was also selected to be relatively small at 125 mm/min. Hence, the tool advance per rotation was 250 μm.

To examine the effect of multiple passes on microstructural evolution, the workpieces with the preplaced Al_2_O_3_ powder were subjected to 1, 2, 3, 4, 8, and 12 FSP passes using the same processing variables as indicated above. In an attempt to promote a more homogeneous distribution of the reinforcing particles, the upper and bottom parts of the workpieces (as well as their advancing and retreating sides) were inverted between the passes, while the processing direction was kept the same.

To assist the interpretation of microstructural changes, the FSP thermal cycle was recorded employing K-type thermocouples placed at the border of the stir zone at the mid-thickness of an FSP workpiece (Figure 2b). The key characteristics of the recorded temperature profile included a peak temperature of 422 °C and a relatively slow cooling rate.

### 2.3. Microstructural Observations

As discussed in Section 1, microstructural observations in the present work were focused on the longitudinal (PD × ND) plane of the stir zone (Figure 1a). The examinations involved optical microscopy, microhardness mapping, and electron backscatter diffraction (EBSD).

In all cases, the FSP’ed workpieces were sectioned in half along the FSP centerline, and metallographic specimens were prepared according to ASTM E3 standards. The final polishing step comprised a long-term (up to 24-h) vibratory polishing with OPS suspension.

The macro-scale structure was studied using optical microscopy and microhardness mapping. The optical observations were conducted using an optical microscope Olympus GX71 equipped with SIAMS 800 software. The Vickers microhardness measurements were performed in accordance with ASTM E92-17 standard by applying a load of 0.2 kg, a dwell time of 10 s, and a step size of 0.25 mm.

The grain structure was investigated by EBSD (the basic principles of EBSD are detailed in Refs. [34,35]) using an FEI Quanta 600 field-emission-gun scanning-electron microscope (SEM) equipped with TSL OIM software. Orientation mapping was conducted employing a scan step size of either 0.5 or 0.25 μm. The low-angle boundaries (LABs) were differentiated from the high-angle boundaries (HABs) using a 15-degree tolerance. The grain size was measured using the equivalent-circle-diameter approach [35].

## 3. Results and Discussion

### 3.1. Macro-Scale Structure

The evolution of the macro-scale structure within the stir zone as a function of the number of FSP passes is shown in Figure 3. To assist the interpretation of the macro-scale structures, microhardness data were also presented in Figure 4 and Figure 5. Specifically, the microhardness maps in Figure 4 were used for the examination of the spatial distribution of Al_3_O_3_ powder. By analogy with the work by Zhang et al. [36], microhardness profiles were also measured to quantify the dispersion of the reinforcing particles through the matrix material (Figure 5a). Moreover, the average microhardness of the acquired microhardness map was also shown as a function of the number of FSP passes in Figure 5b.

After the first FSP pass, the Al_2_O_3_ powder mainly remained in close proximity to the areas of initial placement (indicated by circles in Figure 3a). The spatial dispersion of the reinforcing particles was fairly limited (Figure 3a), thus giving rise to essential variations in microhardness (Figure 4a and Figure 5a). As a result, the microstructure distribution was fairly non-uniform (Figure 3a and Figure 4a). Hence, a single FSP pass is obviously not sufficient to produce a high-quality composite.

Essential particle dispersion in the vertical direction was also worthy of remark (Figure 3a). This observation suggested an extensive vertical material flow within the stir zone.

After two FSP passes, the macro-scale structure became more uniform (compare Figure 3a,b). This resulted in a gradual increase in microhardness (Figure 5b). On the other hand, the reinforcing particles still tended to preferentially concentrate relatively near their initial placement positions (Figure 4b), thereby leading to essential fluctuations in the microhardness profile (Figure 5a).

The third and fourth FSP passes resulted in the development of a specific structure, which consisted of the regular set of vertically oriented particle-rich bands throughout the stir zone (Figure 3c,d). Importantly, the mean spacing between the bands was close to 250 μm, i.e., virtually the tool advance per revolution. Therefore, the developed structure represented the so-called “onion-ring” structure, which is intrinsic to friction stir welding/processing. In fact, this structure represents a superposition of material layers, which are “cut” from the initial workpiece during every single rotation of the FSP tool [37]. In the present case, the development of the “onion-ring” structure implied the formation of the repeated sequence of the particle-rich and particle-poor layers, i.e., virtually, the development of a new sort of microstructural heterogeneity. The preferential arrangement of the reinforcing particles as the onion-ring structure has been reported in the scientific literature [38,39,40,41,42,43,44].

An increased fraction of the reinforcing particles in either the upper or bottom parts of the stir zone was also evident (Figure 3c,d and Figure 4c,d). This observation likely suggested the importance of vertical material flow.

With the further increase in the number of FSP passes, the “onion-ring” structure tended to become less apparent (Figure 3e,f); thus, perhaps evidencing a gradual dispersion of the coarse-scale particle agglomerations. This resulted in a smoothing of microhardness profiles (Figure 5a) and a gradual increase in the average microhardness (Figure 5b). Remarkably, the latter process tended to saturate (Figure 5b).

On the other hand, it is important to emphasize that the agglomerations of Al_2_O_3_ did not disappear completely even after twelve FSP passes (Figure 3f). As a result, the microhardness distribution was still not uniform (Figure 4f). Moreover, the macro-scale cluster of the reinforcing particles at the root of the stir zone also survived after a very large number of FSP passes (Figure 3e,f and Figure 4e).

Thus, even twelve FSP passes are not sufficient to provide an entirely homogeneous distribution of the reinforcing particles.

### 3.2. Grain Structure: FSP without Using Reinforcing Particles

To evaluate the broad aspects of grain structure evolution, a single FSP pass *without* using reinforcing particles was applied. The typical microstructure revealed within the stir zone is shown in Figure 6a.

In accordance with expectations, the microstructure was comprised of comparatively-fine grains (with the average grain size being ~3.5 μm), which contained the essential fraction of LABs. This microstructure is typically revealed in FSP’ed aluminum alloys and is usually attributable to the development of continuous dynamic recrystallization [6,45,46].

Remarkably, the measured HAB fraction (≈80%) was somewhat higher than that achievable during continuous recrystallization (60–70% [47,48]). This observation may be indicative of static grain coarsening, which may occur during the FSP cooling cycle [49], due to the relatively low cooling rate (Figure 2b).

To examine crystallographic texture in the stir zone, the measured orientation data were rotated in order to align them with the local geometry of the simple-shear strain within the stir zone [50]. Specifically, to align the shear direction horizontally, the experimental pole figure was rotated by 90° around the normal direction. Then, to align the shear plane normal vertically, the pole figures were additionally tilted by 60° around the transverse direction. The rotated pole figures were shown in Figure 6b. It is seen that the crystallographic texture within the stir zone was dominated by $B/\bar{B}\{112\}<110>$ simple-shear orientation, which is typically observed in friction-stirred aluminum alloys [49,50].

### 3.3. Effect of the Reinforcing Al_2_O_3_ Nanoscale Particles on Grain-Structure Evolution

#### 3.3.1. Low-Magnification Overview

A series of low-magnification EBSD orientation maps taken from FSP’ed materials are shown in Figure 7. In the maps, grains are colored according to their crystallographic orientations relative to the transverse direction, while black clusters show pixels with a confidence index below 0.1 (which presumably represent the agglomerations of Al_2_O_3_).

It is seen that such clusters are arranged into bands that are oriented nearly vertically (Figure 7a–d), thus resembling the optical microscopy observations (Figure 3). Moreover, in good accordance with optical microscopy, the black clusters tended to disappear with increasing numbers of FSP passes (Figure 7b–f); thus, perhaps evidencing a gradual dispersion of the Al_2_O_3_ particles. Of particular interest was the formation of textural bands after twelve FSP passes (Figure 7f). The possible origin of such bands is considered in Section 3.3.3.

#### 3.3.2. Microstructure Morphology and Grain Size

In this study, no reliable EBSD data were obtained from the coarse-scale particle-rich bands in Figure 7. Hence, microstructural analysis was focused on the aluminum matrix. The high-magnification EBSD maps taken from the aluminum matrix after different numbers of FSP passes are shown in Figure 8. In the maps, LABs and HABs are depicted as red and black lines, respectively. The relevant microstructural statistics are given in Figure 9.

After the first and second FSP passes, the important microstructural characteristic was the development of fine-grained bands (arrows in Figure 8a,b). Remarkably, such bands (as well as the adjacent areas) often exhibited increased LAB content (Figure 8b). These observations presumably reflected the influence of the nanoscale Al_2_O_3_ dispersoids.

Specifically, it is known that the relatively small (~1 μm in size) agglomerations of the nanoscale particles may promote the activation of the mechanism of the particle-stimulated recrystallization [6] and thus promote local grain refinement. On the other hand, the nanoscale dispersoids per se should retard grain-boundary migration and dislocation slip. In the first case, the enhancement of the grain-refinement effect is expected. The second effect may inhibit the progressive evolution of deformation-induced boundaries, including the LAB-to-HAB transformation; this should result in a decreased HAB fraction in the microstructure. It was, therefore, likely that the revealed fine-grained bands were associated with the local concentration of the Al_2_O_3_ particles.

Therefore, the gradual dispersion of the reinforcing particles throughout the aluminum matrix with sequential FSP passes resulted in a steady decrease in both grain size and HAB content (Figure 8c–f and Figure 9a,b). A gradual grain refinement within the matrix of the aluminum-based composites with the number of FSP passes has been reported in the scientific literature [16,17,20,21,51,52,53,54].

Remarkably, both above processes tended to saturate, and thus the final effect was comparatively small (Figure 9). Moreover, a significant experimental scattering is worthy of remarking (Figure 9a,b). The latter effect was likely due to the macro-scale inhomogeneity of microstructure distribution, as discussed above.

#### 3.3.3. Crystallographic Texture

To investigate the evolution of crystallographic texture, orientation data were derived from EBSD maps in Figure 7 and rotated in a manner described in Section 3.2. The typical examples of rotated pole figures are shown in Figure 10. It is seen that neither the addition of the reinforcing particles nor the number of FSP passes exerted a principal influence on the texture. In all cases, it was dominated by $B/\bar{B}\{112\}<110>$ simple-shear orientations.

Considering the formation of textural bands after 12 FSP passes (Figure 7f), the crystallographic orientations of the bands were also examined. To a first approximation, two different types of textural bands could be defined in Figure 7f: (i) the green and (ii) the blue-to-pink colored ones. The crystallographic orientations of the bands are shown in Figure 11.

In both cases, those were close to the $B/\bar{B}\{112\}<110>$ texture components. However, in the green-colored bands, the <110> axis was precisely aligned with the transverse direction (i.e., the local shear direction) (Figure 11a), as is normally expected for FSP-induced texture. In the blue-to-pink colored bands, the <110> axis significantly deviated from this direction (Figure 11b). One of the possible explanations for this observation may be the wobbling of the FSP tool.

## 4. Conclusions

This work was initiated to provide new insight into microstructural evolution during the fabrication of aluminum-matrix composites by FSP. Considering the fact that the reinforcing particles during FSP are usually introduced through the grooves, which are placed along the FSP path, the microstructural examinations in the current study were focused on the longitudinal section of the produced composite. The main conclusions derived from this work are as follows.
(1)Three or four FSP passes, which are often applied in practice, are not sufficient to provide a homogeneous dispersion of the reinforcing particles throughout the matrix material. It was found that the particles rapidly rearranged into the “onion-ring” structure, which was very stable against the subsequent FSP passes. In fact, the remnants of the “onion-ring” structure as well as the comparatively coarse-scale particle agglomerations in the bottom part of the processed material have survived even after 12 FSP passes.(2)The gradual dispersion of the reinforcing particles throughout the aluminum matrix promoted a subtle reduction in grain size and HAB fraction. These observations were suggested to originate from the combined effects of the particle-stimulated recrystallization as well as the retardation of grain-boundary migration and LAB-to-HAB transformation during FSP.


## Figures and Tables

**Figure 1 materials-16-02898-f001:**
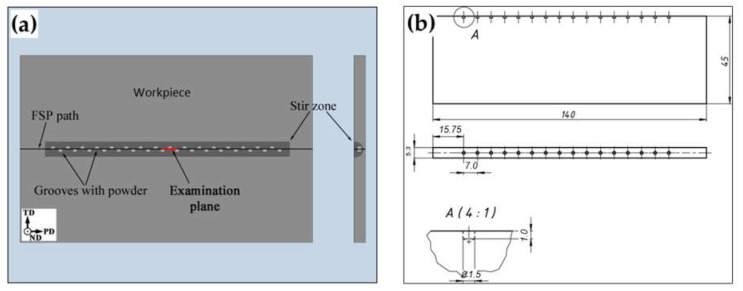
Schematics showing (**a**) the spatial arrangement of the grooves with Al_2_O_3_ powder during FSP and (**b**) the dimensions of the FSP workpieces with drilled grooves. In (**a**), PD, ND, and TD are the FSP direction, normal direction, and transverse direction, respectively.

**Figure 2 materials-16-02898-f002:**
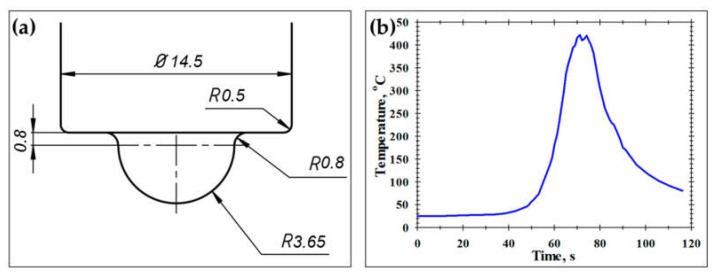
(**a**) Schematic of FSP tool and (**b**) FSP thermal cycle recorded at the border of the stir zone.

**Figure 3 materials-16-02898-f003:**
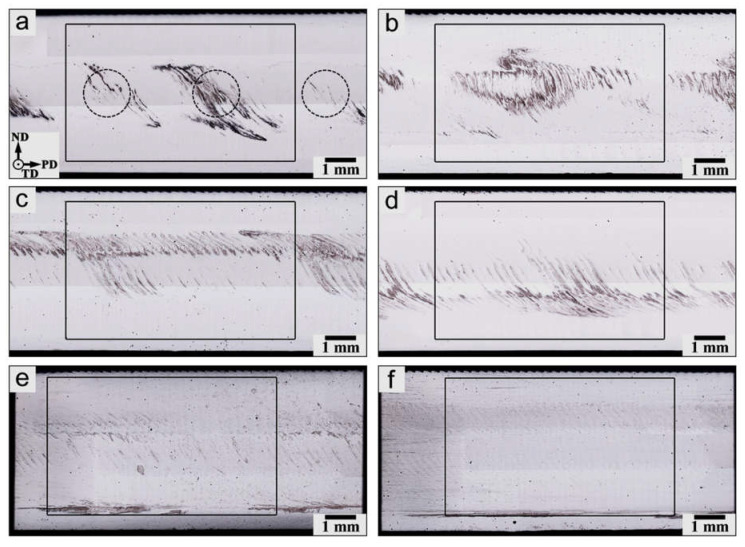
Optical images of longitudinal sections of the processed material as a function of the number of FSP passes: (**a**) 1 pass; (**b**) 2 passes; (**c**) 3 passes; (**d**) 4 passes; (**e**) 8 passes; and (**f**) 12 passes. Selected areas outline microhardness maps shown in Figure 4. In (**a**), circles indicate the approximate positions of the initial placement of alumina powder.

**Figure 4 materials-16-02898-f004:**
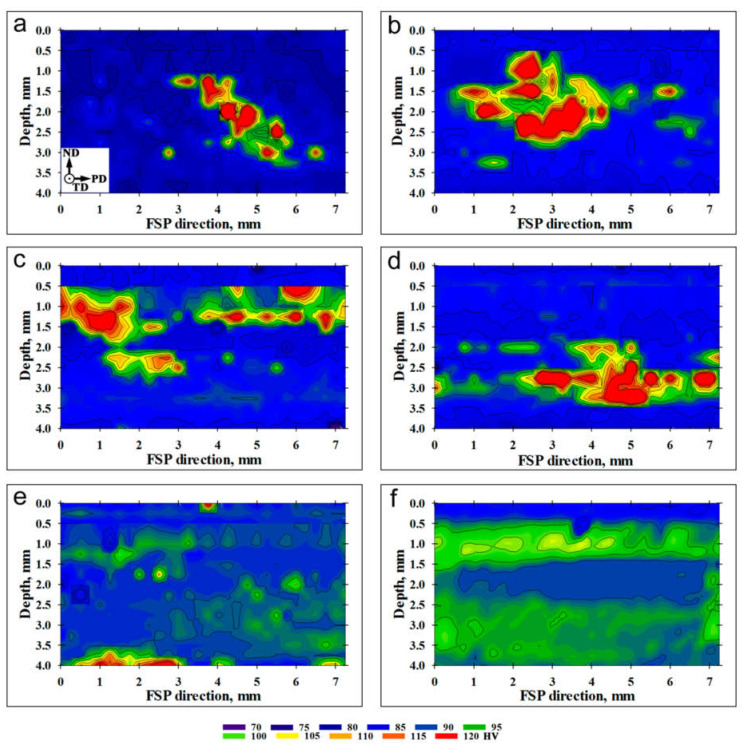
Microhardness maps of the processed material as a function of the number of FSP passes: (**a**) 1 pass; (**b**) 2 passes; (**c**) 3 passes; (**d**) 4 passes; (**e**) 8 passes; and (**f**) 12 passes.

**Figure 5 materials-16-02898-f005:**
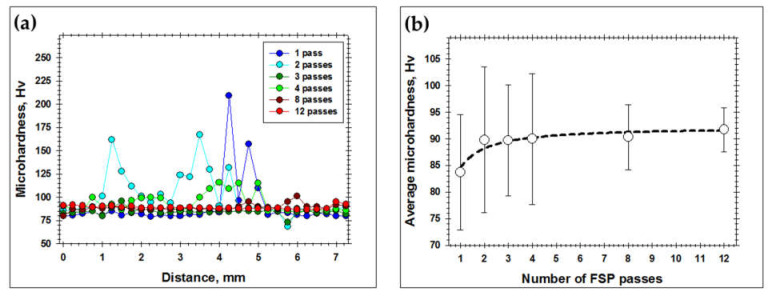
Effect of number FSP passes on (**a**) the microhardness profile measured across the workpieces’ mid-thickness and (**b**) the average microhardness of the acquired microhardness maps. In (**b**), error bars show the standard deviation. Note: The empty circles in (**b**) show experimental data points.

**Figure 6 materials-16-02898-f006:**
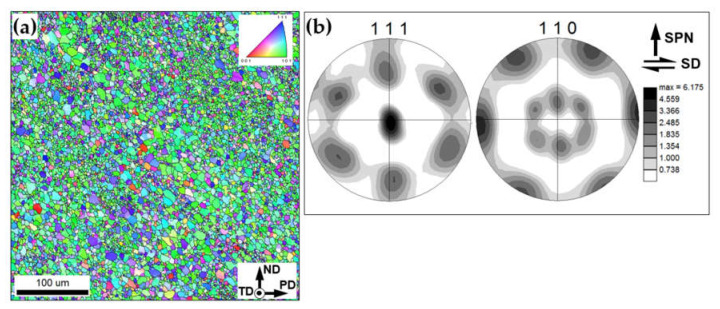
Microstructure produced after a single FSP pass without using Al_2_O_3_ powder: (**a**) EBSD map and (**b**) 111 and 110 pole figures showing crystallographic texture. In (**a**), grains are colored according to the transverse direction; LABs and HABs are depicted as white and black lines, respectively. In (**b**), SD and SPN show the presumed orientations of shear direction and shear plane normal, respectively.

**Figure 7 materials-16-02898-f007:**
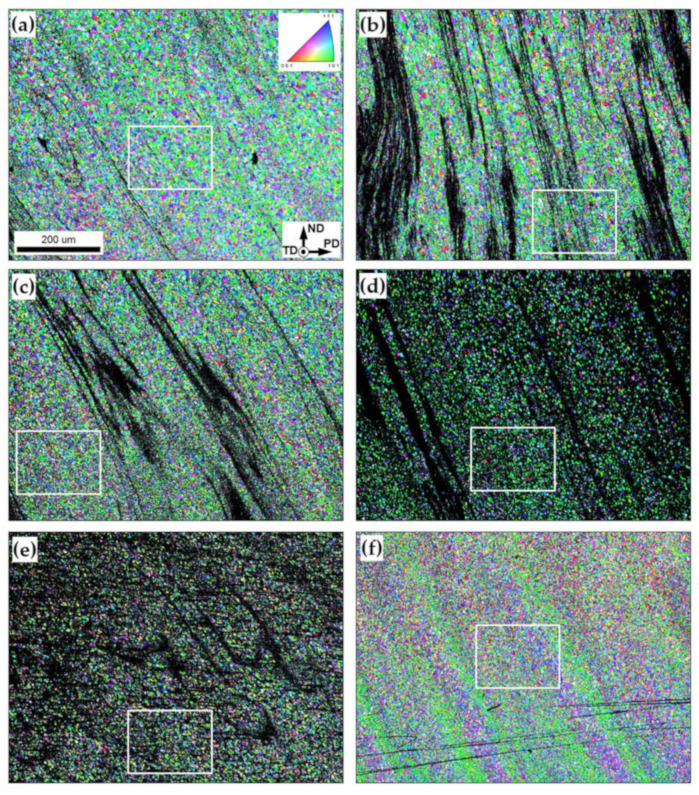
Low-magnification EBSD maps showing grain structures of the produced composite as a function of the number of FSP passes: (**a**) 1 pass; (**b**) 2 passes; (**c**) 3 passes; (**d**) 4 passes; (**e**) 8 passes; and (**f**) 12 passes. In the maps, grains are colored according to their crystallographic orientations relative to the transverse direction, while black clusters show the pixels with a confidence index below 0.1. The reference frame and scale of all EBSD maps are indicated in (**a**). Note: The microstructures within the selected areas (white rectangles) are shown at higher magnification in Figure 7.

**Figure 8 materials-16-02898-f008:**
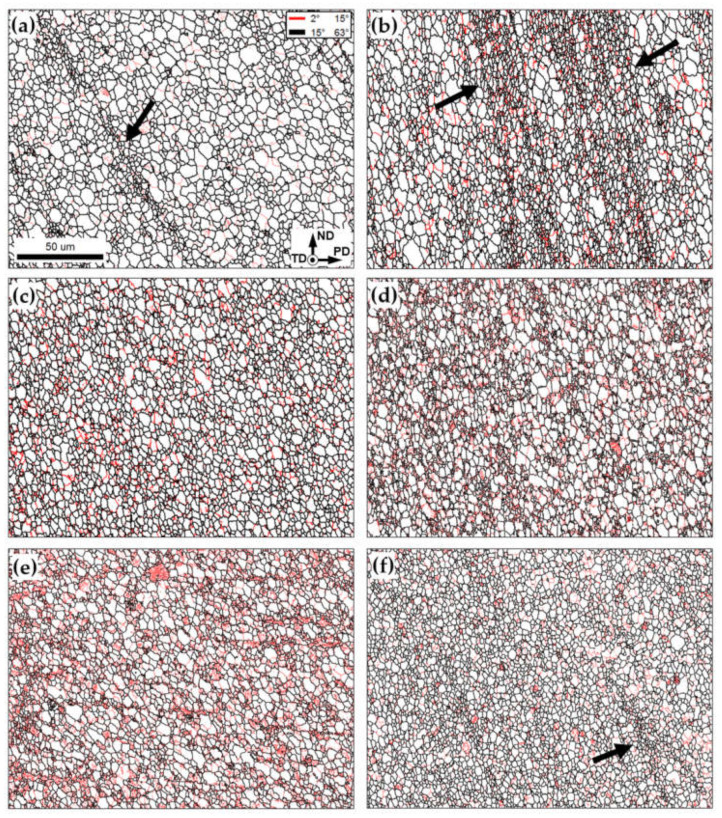
High-magnification EBSD grain-boundary maps showing grain structures within the aluminum matrix as a function of the number of FSP passes: (**a**) 1 pass; (**b**) 2 passes; (**c**) 3 passes; (**d**) 4 passes; (**e**) 8 passes; and (**f**) 12 passes. In the maps, LABs and HABs are depicted as red and black lines, respectively. The reference frame and scale of all EBSD maps are indicated in (**a**). Arrows exemplify the presumed agglomerations of the Al_2_O_3_ powder.

**Figure 9 materials-16-02898-f009:**
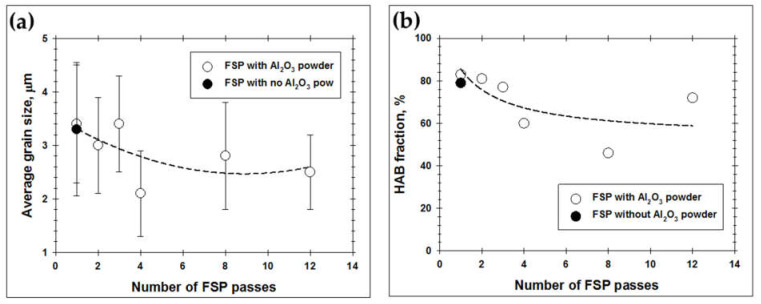
Effect of number FSP passes on (**a**) average grain size and (**b**) HAB fraction within the aluminum matrix of the produced composite.

**Figure 10 materials-16-02898-f010:**
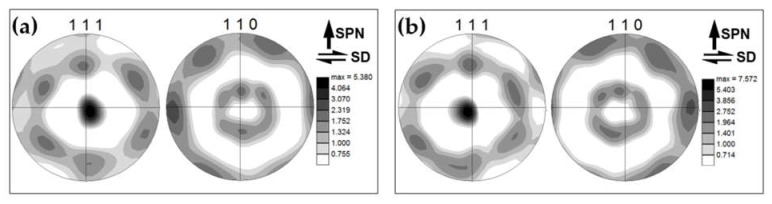
Images of 111 and 110 pole figures showing crystallographic texture within the aluminum matrix of the produced composite as a function of the number of FSP passes: (**a**) 1 pass; (**b**) 12 passes. SD and SPN show the presumed orientations of shear direction and shear plane normal, respectively.

**Figure 11 materials-16-02898-f011:**
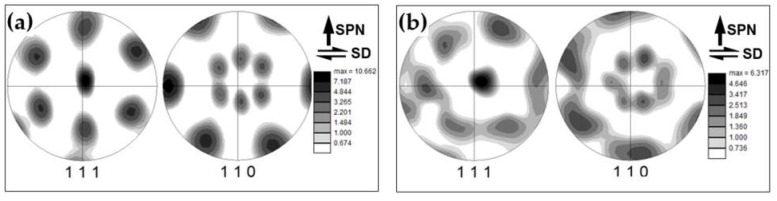
Images of 111 and 110 pole figures showing crystallographic orientations of textural bands in the material condition produced after 12 FSP passes (Figure 7f): (**a**) green-colored bands; and (**b**) blue-to-pink colored bands. SD and SPN shows the presumed orientations of shear direction and shear plane normal, respectively.

## Data Availability

The data presented in this study are available on request from the corresponding author.

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
