# Peer review of "Microstructural Aspects of the Fabrication of Al/Al2O3 Composite by Friction Stir Processing"

_materials, 2023, doi:10.3390/ma16072898_

Round 1
Reviewer 1 Report
Journal: Materials (ISSN 1996-1944)
Manuscript ID: materials-2303515
Review Report 1#
The authors presented an article on “Microstructural aspects of the fabrication of Al/Al2O3 composite by friction stir processing”. The subject of the article falls within the scope of the journal "Materials". However, the article will be ready for publication after a major revision. Comments are listed below.
1. The similarity rate is 37%. It should be reduced.
2. The introduction part is very inadequate. It should definitely be expanded. In the introduction, the usage areas of aluminum composites and the difference from other composites should be mentioned. Also, references should be increased.
3. There are many studies on Al/Al2O3 composites. What's new in this article? The innovative aspect of this study should be emphasized in the last paragraph of the introduction.
4. On page 2, line 63, "Fig 1a" should be "Fig 2a".
5. According to which standards were the microstructure and microhardness analyses performed?
6. The production stages of the composite should be detailed in the material and method section. For example, the chemical composition of the Al5182 alloy can be given.
7. The label of Figure 3-4-5-6-9-10 are too long. It should be shortened.
8. In Figure 3, the expression "(d) 12 passes" should be "(f) 12 passes".
9. In this study, the focus is on the microstructural analysis of the composite. However, it is seen that there is no X-RD analysis. X-RD analysis is needed to determine the phases and compounds formed in the produced composite.
10. In Figure 4, the expression "(d) 12 passes" should be "(f) 12 passes".
11. When Figure 4 is examined, it is seen that the microhardness decreases as the FSP passes increase. The reasons for this should be explained.
12. What is the reason for the fluctuation in average grain size as the number of FSP passes increases in Figure 8? It should be explained.
13. The Results section is generally devoid of discussion. Please discuss the results using similar previous studies.
14. On page 10, line 225, "Summary" should be "Conclusions".
15. The article contains numerous typographic and language errors. It should be corrected.
16. The article should be rearranged by taking into account the journal writing rules and citation rules.
17. Your reference list contains only one article from the "Materials" journal. If your work is appropriate for the context of this journal, there are many references from this journal. Second, the cited sources should be primary sources. So the indexed area shows the strength of a paper and directly the reliability of your paper. Please make arrangements in this direction.
*** Authors must consider them properly before submitting the revised manuscript. A point-by-point reply is required when the revised files are submitted.

Author Response
Responses to Reviewers’ comments on the paper entitled
“Microstructural aspects of the fabrication of Al/Al2O3 composite by friction stir processing” (Ms. Ref. No.: materials-2303515)
The authors would like to express their gratitude to Reviewer for useful remarks to improve the paper. Below, we provided specific replies to the issues raised.
The authors presented an article on “Microstructural aspects of the fabrication of Al/Al2O3 composite by friction stir processing”. The subject of the article falls within the scope of the journal "Materials". However, the article will be ready for publication after a major revision. Comments are listed below.
- The similarity rate is 37%. It should be reduced.
Authors’ response
The authors carefully checked the duplication report provided by “Materials”. We found that a significant portion of the duplicated text (808 out of 1184 words) was associated with the own authors’ articles. Hence, the increased similarity index was perhaps not indicative of plagiarism. Since the authors are not the native English persons, our lexicon is limited. Thus, we often use some typical word constructions in writing. On the other hand, the authors would like to emphasize that the idea, results, their interpretations, and conclusions of this work are completely original.
Nevertheless, to address the Reviewer’s comment, the authors did their best in the rephrasing of the manuscript. Accordingly, appropriate changes have been made throughout the text of the revised paper.
- The introduction part is very inadequate. It should definitely be expanded. In the introduction, the usage areas of aluminum composites and the difference from other composites should be mentioned. Also, references should be increased.
Authors’ response
According to the comment, the Introduction part has been carefully revised. Specifically, it has been expanded, and particular emphasis has been paid to the consideration of the recommended issues.
The revised Introduction is shown below Pages 1 to 2, Lines 26 to 57):
“Aluminum matrix composites offer an attractive synergy of properties, including low density, good strength, and superior thermal stability [1]. Accordingly, these materials are used for high-performance applications in the transportation sector (and, particularly, in the automotive industry), specifically, for the manufacturing of pistons, brake rotors, connecting rods, etc. [2]. An important advantage of aluminum matrix composites if compared with other lightweight composites (such as magnesium or titanium-based) is a reasonably low price-performance ratio [3].
Friction stir processing (FSP) is sometimes considered a promising technique for the fabrication of aluminum matrix composites [4-7]. Due to the solid-state nature of this technology, FSP minimizes interfacial reactions between the reinforcing elements and the aluminum matrix and thus provides good interfacial bonding between these two constituents.
The extensive research in this area has conclusively demonstrated the high efficiency of FSP for the fabrication of aluminum matrix composites [4-7]. Specifically, a wide range of composites was successfully produced using a variety of both reinforcing particles and matrix aluminum alloys. The reinforcing elements typically comprise SiC [8-15] or Al2O3 [11, 16-25] phases but may also involve other particles [26-33]. On the other hand, the matrix materials included the 1xxx [8,9,16,29], 2xxx [20-22], 5xxx [11,13,27,30,32,33], 6xxx [12,17,23,31], and 7xxx [10,26] series of aluminum alloys.
On the other hand, it was found that the agglomeration of the reinforcing particles during FSP represents an essential problem. Specifically, a single FSP pass is usually not sufficient to produce a homogeneous particle distribution. Typically, three to four passes are applied, therefore [11,17-21].
It is important to point out that the distribution of the reinforcing particles throughout the FSP-produced composites is normally examined in the transverse cross section of the processed material. In this context it is worth noting that the FSP technology for composite fabrication is based on the preplaced powder strategy. According to this technique, the reinforcing particles (or powder) are distributed throughout the matrix material from the preliminary-drilled grooves, which are preplaced along the FSP path. Hence, it would be useful to also check the particle distribution in the longitudinal section of the composites. This approach may provide new insight into microstructural evolution during the FSP fabrication of aluminum matrix composites and thus improve our understanding of this process. This was the objective of the present work.”
The new references added to the revised manuscript are shown below:
"2. Mavhungu, S.T.; Akinlabi, E.T.; Onitiri, M.A.; Varachia, F.M. Aluminum Matrix Composites for Industrial Use: Advances and Trends. Proc. Manuf. 2017, 7, 178-182. https://doi.org/10.1016/j.promfg.2016.12.045.
- Torralba, J.M.; Costa, C.E.; Velasco, F. P/M Aluminum Matrix Composites: An Overview. J. Mater. Proc. Technol. 2003, 133, 203-206. https://doi.org/10.1016/S0924-0136(02)00234-0.
- Gan, Y.X.; Solomon, D.; Reinbolt, M. Friction Stir Processing of Particle Reinforced Composite Materials. Materials2010, 3, 329-350. https://doi.org/10.3390/ma3010329.
- Huang, C.W.; Aoh, J.N. Friction Stir Processing of Copper-Coated SiC Particulate-Reinforced Aluminum Matrix Composite. Materials2018, 11, 599. https://doi.org/10.3390/ma11040599.
- Khodabakhshi, F.; Nosko, M.; Gerlich, A.P. Dynamic Restoration and Crystallographic Texture of a Friction-Stir Processed Al-Mg-SiC Surface Nanocomposite. Mater. Sci. Tech. 2018, 34, 1773-1791. https://doi.org/10.1080/02670836.2018.1490858
- Wojcicka, A.; Mroczka, K.; Kurtyka, P.; Binkowski, M.; Wrobel, Z. X-ray Microtomography Analysis of the Aluminum Alloy Composite Reinforced by SiC after Friction Stir Processing. J. Mater. Eng. Perform. 2014, 23, 3215-3221. https://doi.org/10.1007/s11665-014-1097-2.
- Wang, W.; Shi, Q.; Liu, P.; Li, H.; Li, T. A Novel Way to Produce Bulk SiCp Reinforced Aluminum Metal Matrix Composites by Friction Stir Processing. J. Mater. Proc. Technol. 2009, 209, 2099-2103. https://doi.org/10.1016/j.matprotec.2008.05.001.
- Moustafa, E.B. Dynamic Characteristics Study for Surface Composite of AMMNCs Matrix Fabricated by Friction Stir Process. Materials2018, 11, 1240. https://doi.org/10.3390/ma11071240.
- Shafiei-Zarghani, A.; Kashani-Bozorg, S.F.; Zarei-Hanzaki, A. Microstructures and Mechanical Properties of Al/Al2O3 Surface Nano-Composite Layer Produced by Friction Stir Processing, Mater. Sci. Eng. A 2009, 500, 84-91. https://doi.org/10.1016/j.msea.2008.09.064.
- Mazaheri, Y.; Karimzadeh, F.; Enayati, M.H. A Novel Technique for Development of A356/Al2O3 Surface Nanocomposite by Friction Stir Processing, J. Mater. Proc. Technol. 2011, 211, 1614-1619. https://doi.org/10.1016/j.jmatprotec.2011.04.015.
- Mazaheri, Y., Karimzadeh, F.; Enayati, M.H. Tribological Behavior of A356/Al2O3 Surface Nanocomposite Prepared by Friction Stir Processing. Metall. Mater. Trans. A 2014, 45, 2250–2259. https://doi.org/10.1007/s11661-013-2140-x.
- Mahmoud E.R.I.; Tash M.M. Characterization of Aluminum-Based-Surface Matrix Composites with Iron and Iron Oxide Fabricated by Friction Stir Processing. Materials (Basel) 2016, 23, 505. doi: 10.3390/ma9070505.
- 30. Papantoniou, I.G.; Markopoulos, A.P.; Manolakos, D.E. A New Approach in Surface Modification and Surface Hardening of Aluminum Alloys Using Friction Stir Process: Cu-Reinforced AA5083. Materials2020, 13, 1278. https://doi.org/10.3390/ma13061278.
- Guo, L.; Liu, Y.; Shen, K.; Song, C.; Yang, M.; Kim, K.; Wang, W. Enhancing Corrosion and Wear Resistance of AA6061 by Friction Stir Processing with Fe78Si9B13Glass Particles. Materials 2015, 8, 5084-5097. https://doi.org/10.3390/ma8085084.
- Rubtsov, V.; Chumaevskii, A.; Gusarova, A.; Knyazhev, E.; Gurianov, D.; Zykova, A.; Kalashnikova, T.; Cheremnov, A.; Savchenko, N.; Vorontsov, A.; Utyaganova, V.; Kolubaev, E.; Tarasov, S. Macro- and Microstructure of In-Situ Composites Prepared by Friction Stir Processing of AA5056 Admixed with Copper Powders. Materials2023, 16, 1070. https://doi.org/10.3390/ma16031070.
- Chumaevskii, A.; Zykova, A.; Sudarikov, A.; Knyazhev, E.; Savchenko, N.; Gubanov, A.; Moskvichev, E.; Gurianov, D.; Nikolaeva, A.; Vorontsov, A.; Kolubaev, E.; Tarasov, S. In-Situ Al-Mg Alloy Base Composite Reinforced by Oxides and Intermetallic Compounds Resulted from Decomposition of ZrW2O8during Multipass Friction Stir Processing. Materials 2023, 16, 817. https://doi.org/10.3390/ma16020817.
- Adams, B.L.; Wright, S.I.; Kunze, K. Orientation imaging: The Emergence of a New Microscopy. Metal. Tans. A 1993, 24, 819-831. https://doi.org/10.1007/BF02656503.
- Humphreys, F.J. Quantitative Metallography by Electron Back-Scattered Diffraction. J. Microsc. 1999, 195, 170-185. https://doi.org/10.1046/j.1365-2818.1999.00578.x.
- Zhang, C.; Hu, X.; Lu, P. Fatigue and Hardness Effects of a Thin Buffer Layer on the Heat Affected Zone of a Weld Repaired Bisplate80, J. Mater. Proc. Technol. 2012, 212, 393-401. https://doi.org/10.1016/j.jmatprotec.2011.10.002.
- Abraham, S.J.; Dinaharan, I.; Selvam, J.D.R.; Akinlabi, E.T. Microstructural Characterization of Vanadium Particles Reinforced AA6063 Aluminum Matrix Composites via Friction Stir Processing with Improved Tensile Strength and Appreciable Ductility. Composites Communications 2019, 12, 54-58. https://doi.org/10.1016/j.coco.2018.12.011.
- Khan, M.; Rehman, A.; Aziz, T.; Naveed, K.; Ahmad, I.; Subhani, T. Cold Formability of Friction Stir Processed Aluminum Composites Containing Carbon Nanotubes and Boron Carbide Particles. Mater. Sci. Eng. A 2017, 701, 382-388. https://doi.org/10.1016/j.msea.2017.05.121.
- Arab, S.M.; Karimi, S.; Jahromi, S.A.J.; Javadpour, S.; Zebarjad, S.M. Fabrication of Novel Fiber Reinforced Aluminum Composites by Friction Stir Processing. Mater. Sci. Eng. A 2015, 632, 50-57. https://doi.org/10.1016/j.msea.2015.02.032.
- Jeon, C.H.; Jeong, Y.H.; Seo, J.J.; Tien, H.N.; Hong, S.T.; Yum, Y.J.; Hur, S.H.; Lee, K.J. Material Properties of Graphene/Aluminum Metal Matrix Composites Fabricated by Friction Stir Processing. Int. J. Precis. Eng. Manuf.2014, 15, 1235–1239. https://doi.org/10.1007/s12541-014-0462-2.
- Liu, Q.; Ke, L.; Liu, F.; Huang, C.; Xing, L. Microstructure and Mechanical Property of Multi-Walled Carbon Nanotubes Reinforced Aluminum Matrix Composites Fabricated by Friction Stir Processing, Mater. Design 2013, 45, 343-348. https://doi.org/10.1016/j.matdes.2012.08.036.
- Lee, I.S.; Hsu, C.J.; Chen, C.F.; Ho, N.J.; Kao, P.W. Particle-Reinforced Aluminum Matrix Composites Produced from Powder Mixtures via Friction Stir Processing. Composit. Sci. Technol. 2011, 71, 693-698. https://doi.org/10.1016/j.compscitech.2011.01.013.
- Wang, W.; Shi, Q.; Liu, P.; Li, H.; Li, T. A Novel Way to Produce Bulk SiCp Reinforced Aluminum Metal Matrix Composites by Friction Stir Processing. J. Mater. Proc. Technol. 2009, 209, 2099-2103. https://doi.org/10.1016/j.jmatprotec.2008.05.001.
- Moustafa, E.B.; AbuShanab, W.S.; Ghandourah, E.; Taha, M.A. Microstructural, Mechanical and Thermal Properties Evaluation of AA6061/Al2O3-BN Hybrid and Mono Nanocomposite Surface. J. Mater. Res. Technol. 2020, 9, 15486-15495. https://doi.org/10.1016/j.jmrt.2020.11.010.
- Moustafa, E.B.; Melaibari, A.; Alsoruji, G.; Khalil, A.M.; Mosleh, A.O. Tribological and Mechanical Characteristics of AA5083 Alloy Reinforced by Hybridising Heavy Ceramic Particles Ta2C & VC with Light GNP and Al2O3 Nanoparticles, Ceramic Inter. 2022, 48, 4710-4721. https://doi.org/10.1016/j.ceramint.2021.11.007.
- Moustafa, E.B.; Abushanab, W.S.; Melaibari, A.; Yakovtseva, O.; Mosleh, A.O. The Effectiveness of Incorporating Hybrid Reinforcement Nanoparticles in the Enhancement of the Tribological Behavior of Aluminum Metal Matrix Composites. JOM2021, 73, 4338–4348. https://doi.org/10.1007/s11837-021-04955-w.
- Abushanab, W.S.; Moustafa, E.B.; Melaibari, A.A.; Kotov, A.D.; Mosleh, A.O. A Novel Comparative Study Based on the Economic Feasibility of the Ceramic Nanoparticles Role’s in Improving the Properties of the AA5250 Nanocomposites. Coatings2021, 11, 977. https://doi.org/10.3390/coatings11080977.”
- There are many studies on Al/Al2O3 composites. What's new in this article? The innovative aspect of this study should be emphasized in the last paragraph of the introduction.
Authors’ response
The main idea of the present work is the microstructural characterization of the FSP produced composite in its longitudinal section. This idea is based on the fact that the reinforcing particles during FSP are distributed throughout the matrix material from the preliminary-drilled grooves, which are preplaced along the FSP pass. Hence, their inhomogeneous distribution should be expected in the longitudinal section of the FSP’ed material. On the other hand, microstructural observations of the FSP-produced composites are typically focused on their transverse cross-section. The authors believe that this approach is not entirely straightforward for microstructural characterization. Hence, the present study was undertaken with the hope of providing a new insight into this issue.
According to the comment, the innovative aspect of this study has been emphasized in the last paragraph of the Introduction section (Page 2, Lines 48 to 57):
“It is important to point out that the distribution of the reinforcing particles throughout the FSP-produced composites is normally examined in the transverse cross section of the processed material. In this context it is worth noting that the FSP technology for composite fabrication is based on the preplaced powder strategy. According to this technique, the reinforcing particles (or powder) are distributed throughout the matrix material from the preliminary-drilled grooves, which are preplaced along the FSP path. Hence, it would be useful to also check the particle distribution in the longitudinal section of the composites. This approach may provide new insight into microstructural evolution during the FSP fabrication of aluminum matrix composites and thus improve our understanding of this process. This was the objective of the present work.”
- On page 2, line 63, "Fig 1a" should be "Fig 2a".
Authors’ response
To avoid misunderstanding, the FSP reference frame has been shown in Fig.1a of the revised manuscript (Please see attached file).
- According to which standards were the microstructure and microhardness analyses performed?
Authors’ response
According to the comment, the standards have been indicated in the revised manuscript (Page 3, Section 2.3, Lines 105 and 111):
“In all cases, the FSP’ed workpieces were sectioned in half along the FSP centerline and metallographic specimens were prepared according to ASTM E3 standard. The final polishing step comprised a long-term (up to 24-hrs.) vibratory polishing with OPS suspension.
The macro-scale structure was studied using optical microscopy and microhardness mapping. The optical observations were conducted using optical microscope Olympus GX71 equipped with SIAMS 800 software. The Vickers microhardness measurements were performed in accordance with ASTM E92-17 standard by applying a load of 0.2 kg, a dwell time of 10 s, and a step size of 0.25 mm.”
To the best of the authors’ knowledge, electron backscatter diffraction (EBSD), which was one of the key characterization techniques in the present study, is not regulated by any standard at present. Hence, the classical works, which described the basal principles of EBSD, have been indicated instead (Page 4, footnote):
“The basic principles of EBSD are detailed in Refs. [34,35].”
- Adams, B.L.; Wright, S.I.; Kunze, K. Orientation imaging: The Emergence of a New Microscopy. Metal. Tans. A 1993, 24, 819-831. https://doi.org/10.1007/BF02656503.
- Humphreys, F.J. Quantitative Metallography by Electron Back-Scattered Diffraction. J. Microsc. 1999, 195, 170-185. https://doi.org/10.1046/j.1365-2818.1999.00578.x.
- The production stages of the composite should be detailed in the material and method section. For example, the chemical composition of the Al5182 alloy can be given.
Authors’ response
The section “Materials and Methods” has been revised according to the comment. Specifically, the chemical composition of the aluminum matrix material has been given (Page 2, Section 2.1, Lines 63 to 64):
“The commercial aluminum alloy 5182 was employed as a metal matrix material. The material was manufactured by semi-continuous casting using the casting equipment of the Joint Research Center “Technology and Materials” at Belgorod National Research University. The measured chemical composition of the matrix material was Al-4.75Mg-0.3Mn-0.15Zr-0.1Ti-0.1Cr-0.06Si-0.05Fe (wt. %)… “
- The label of Figure 3-4-5-6-9-10 are too long. It should be shortened.
Authors’ response
According to the comment, the figure captions of Figs. 3, 4, 5, 6, 9, and 10 have been shortened in the revised manuscript.
- In Figure 3, the expression "(d) 12 passes" should be "(f) 12 passes".
Authors’ response
Figure 3 has been revised according to the comment.
- In this study, the focus is on the microstructural analysis of the composite. However, it is seen that there is no X-RD analysis. X-RD analysis is needed to determine the phases and compounds formed in the produced composite.
Authors’ response
Unfortunately, our x-ray measurements were not effective to detect the presence of Al2O3 phase within the produced composite. This was likely due to (i) relatively small content of the reinforcing phase (~1 vol.%) as well as (ii) the comparatively low intensity of x-ray signal from the Al2O3 phase (please see experimental data in the attached version of the response letter).
To avoid misunderstanding, the comparatively low content of the reinforcing phase within the composite has been emphasized in the revised manuscript (Page 2, Lines 73-76):
“…The approximate volume fraction of the alumina powder was evaluated using the equation [9] , where is the total volume of the grooves, is the cross-section area of the stir zone, and is the length of the FSP path (Fig. 1a). The calculated volume fraction was ≈1%.”
- In Figure 4, the expression "(d) 12 passes" should be "(f) 12 passes".
Authors’ response
Figure 4 has been revised according to the comment.
- When Figure 4 is examined, it is seen that the microhardness decreases as the FSP passes increase. The reasons for this should be explained.
Authors’ response
This observation was an artifact associated with inhomogeneous microstructure distribution after low FSP strain. Due to the preferential concentration of Al2O3 powder in selected locations, those exhibited increased microhardness. With an increase in the number of FSP passes, the reinforcing particles gradually distribute throughout the matrix material, thus increasing the average microhardness while decreasing its oscillations.
To emphasize this issue, appropriate microhardness profiles have been measured. Moreover, the evolution of the average microhardness (over the entire dataset) as a function of the number of FSP passes has been calculated. The new results have been arranged as Fig. 5 in the revised manuscript (please see experimental data in the attached file of the response letter).
Accordingly, appropriate discussion has been added to the revised manuscript (Section 3.1, Page 6, Lines 158 to 162)):
“With the further increase in number of FSP passes, the “onion-ring” structure tended to become less apparent (Figs. 3e and f), thus perhaps evidencing a gradual dispersion of the coarse-scale particle agglomerations. This resulted in a smoothing of microhardness profiles (Fig. 5a) and a gradual increase in the average microhardness (Fig. 5b)…”
- What is the reason for the fluctuation in average grain size as the number of FSP passes increases in Figure 8? It should be explained.
Authors’ response
The authors believe that such fluctuations are the result of inhomogeneous microstructure distribution. To avoid misunderstanding, this issue has been discussed in the revised manuscript (Section 3.3.2, Page 10, Lines 228 to 230):
“Also, a significant experimental scattering is worthy of remarking (Figs. 9a and b). The latter effect was likely due to the macro-scale inhomogeneity of microstructure distribution, as discussed above. “
- The Results section is generally devoid of discussion. Please discuss the results using similar previous studies.
Authors’ response
According to the comment, the experimental results have been discussed in the context of scientific literature. This discussion was focused on two principal findings: (i) the development of the onion-ring structure, and (ii) grain refinement associated with the dispersion of nanoscale particles:
Section 3.1, Page 6, Lines 143 to 154:
“The third and fourth FSP passes resulted in the development of a specific structure, which consisted of the regular set of vertically oriented particle-rich bands throughout the stir zone (Figs. 3c and 3d). Importantly, the mean spacing between the bands was close to 250 m, i.e., virtually the tool advance per revolution. Therefore, the developed structure represented the so-called “onion-ring” structure, which is intrinsic to friction stir welding/processing. In fact, this structure represents a superposition of material layers, which are “cut” from the initial workpiece during every single rotation of the FSP tool [37]. In the present case, the development of the “onion-ring” structure implied the formation of the repeated sequence of the particle-rich and particle-poor layers, i.e., virtually, the development of a new sort of microstructural heterogeneity. The preferential arrangement of the reinforcing particles as the onion-ring structure has been reported in the scientific literature [38-44].”
References:
- Krishnan, K.N. On the Formation of Onion Rings in Friction Stir Welds. Mater. Sci. Eng. A 2002, 327, 246-251. doi:10.1016/S0921-5093(o1)01474-5.
- Abraham, S.J.; Dinaharan, I.; Selvam, J.D.R.; Akinlabi, E.T. Microstructural Characterization of Vanadium Particles Reinforced AA6063 Aluminum Matrix Composites via Friction Stir Processing with Improved Tensile Strength and Appreciable Ductility. Composites Communications 2019, 12, 54-58. https://doi.org/10.1016/j.coco.2018.12.011.
- Khan, M.; Rehman, A.; Aziz, T.; Naveed, K.; Ahmad, I.; Subhani, T. Cold Formability of Friction Stir Processed Aluminum Composites Containing Carbon Nanotubes and Boron Carbide Particles. Mater. Sci. Eng. A 2017, 701, 382-388. https://doi.org/10.1016/j.msea.2017.05.121.
- Arab, S.M.; Karimi, S.; Jahromi, S.A.J.; Javadpour, S.; Zebarjad, S.M. Fabrication of Novel Fiber Reinforced Aluminum Composites by Friction Stir Processing. Mater. Sci. Eng. A 2015, 632, 50-57. https://doi.org/10.1016/j.msea.2015.02.032.
- Jeon, C.H.; Jeong, Y.H.; Seo, J.J.; Tien, H.N.; Hong, S.T.; Yum, Y.J.; Hur, S.H.; Lee, K.J. Material Properties of Graphene/Aluminum Metal Matrix Composites Fabricated by Friction Stir Processing. Int. J. Precis. Eng. Manuf.2014, 15, 1235–1239. https://doi.org/10.1007/s12541-014-0462-2.
- Liu, Q.; Ke, L.; Liu, F.; Huang, C.; Xing, L. Microstructure and Mechanical Property of Multi-Walled Carbon Nanotubes Reinforced Aluminum Matrix Composites Fabricated by Friction Stir Processing, Mater. Design 2013, 45, 343-348. https://doi.org/10.1016/j.matdes.2012.08.036.
- Lee, I.S.; Hsu, C.J.; Chen, C.F.; Ho, N.J.; Kao, P.W. Particle-Reinforced Aluminum Matrix Composites Produced from Powder Mixtures via Friction Stir Processing. Composit. Sci. Technol. 2011, 71, 693-698. https://doi.org/10.1016/j.compscitech.2011.01.013.
- Wang, W.; Shi, Q.; Liu, P.; Li, H.; Li, T. A Novel Way to Produce Bulk SiCp Reinforced Aluminum Metal Matrix Composites by Friction Stir Processing. J. Mater. Proc. Technol. 2009, 209, 2099-2103. https://doi.org/10.1016/j.jmatprotec.2008.05.001.
Section 3.3.2, Pages 8o 10, Lines 209 to 226:
“After the first and second FSP passes, the important microstructural characteristic was the development of fine-grained bands (arrows in Fig. 8a and b). Remarkably, such bands (as well as the adjacent areas) often exhibited increased LAB content (Fig. 8b). These observations presumably reflected the influence of the nanoscale Al2O3 dispersoids.
Specifically, it is known that the relatively small (~1 mm in size) agglomerations of the nanoscale particles may promote the activation of the mechanism of the particle-stimulated recrystallization [6] and thus promote local grain refinement. On the other hand, the nano-scale dispersoids per se should retard grain-boundary migration and dislocation slip. In the first case, the enhancement of the grain-refinement effect is expected. The second effect may inhibit the progressive evolution of deformation-induced boundaries, including the LAB-to-HAB transformation; this should result in the decreased HAB fraction in the microstructure. It was therefore likely that the revealed fine-grained bands were associated with the local concentration of the Al2O3 particles.
Therefore, the gradual dispersion of the reinforcing particles throughout the aluminum matrix with sequential FSP passes resulted in a steady decrease in both grain size and HAB content (Figs. 8c-f and 9a-b). A gradual grain refinement within the matrix of the aluminum-based composites with the number of FSP passes has been reported in the scientific literature [16,17,20,21, 51-55].”
References:
- Heidarzadeh, A.; Mironov, S.; Kaibyshev, R.; Cam, G.; Simar, A.; Gerlich, A.P.; Khodabakhshi, A.F.; Mostafaei, A.; Field, D.P.; Robson, J.D.; Deschamps, A.; Withers, P.J. Friction Stir Welding/Processing of Metals and Alloys: A Comprehensive Review on Microstructural Evolution. Prog. Mater. Sci. 2021, 117, 100752. doi:10.1016/j.pmatsci.2020.100752.
- Orlowska, M.; Pixner, F.; Hutter, A.; Enzinger, N.; Olejnik, L.; Lewandowska, M. Manufacturing of Coarse and Ultrafine-Grained Aluminum Matrix Composites Reinforced with Al2O3 Nanoparticles via Friction Stir Processing. J. Manuf. Proc. 2022, 80, 359-373. doi:10.1016/j.jmapro.2022.06011.
- Shafiei-Zarghani, A.; Kashani-Bozorg, S.F.; Zarei-Hanzaki, A. Wear Assessment of Al/Al2O3 Nanocomposite Surface Layer Produced Using Friction Stir Processing. Wear. 2011, 270, 403-412. doi://10.1016/j.weqr.2010.12.002.
- Moustafa, E. Effect of Multi-Pass Friction Stir Processing on Mechanical Properties for AA2024/Al2O3 Nanocomposites. Materials. 2017, 10, 1053. doi:10.3390/ma10091053.
- AbuShanab, W.S.; Moustafa, E.B. Effects of Friction Stir Processing Parameters on the Wear Resistance and Mechanuical Properties of Fabricated Metal Matrix Nanocomposites (MMNCs) Surface. J. Mater. Res. Technol. 2020, 9, 7460-7471. doi:10.1016/j.jmrt.2020.04.073.
- Moustafa, E.B.; AbuShanab, W.S.; Ghandourah, E.; Taha, M.A. Microstructural, Mechanical and Thermal Properties Evaluation of AA6061/Al2O3-BN Hybrid and Mono Nanocomposite Surface. J. Mater. Res. Technol. 2020, 9, 15486-15495. https://doi.org/10.1016/j.jmrt.2020.11.010.
- Moustafa, E.B.; Melaibari, A.; Alsoruji, G.; Khalil, A.M.; Mosleh, A.O. Tribological and Mechanical Characteristics of AA5083 Alloy Reinforced by Hybridising Heavy Ceramic Particles Ta2C & VC with Light GNP and Al2O3 Nanoparticles, Ceramic Inter. 2022, 48, 4710-4721. https://doi.org/10.1016/j.ceramint.2021.11.007.
- Moustafa, E.B.; Abushanab, W.S.; Melaibari, A.; Yakovtseva, O.; Mosleh, A.O. The Effectiveness of Incorporating Hybrid Reinforcement Nanoparticles in the Enhancement of the Tribological Behavior of Aluminum Metal Matrix Composites. JOM2021, 73, 4338–4348. https://doi.org/10.1007/s11837-021-04955-w.
- Abushanab, W.S.; Moustafa, E.B.; Melaibari, A.A.; Kotov, A.D.; Mosleh, A.O. A Novel Comparative Study Based on the Economic Feasibility of the Ceramic Nanoparticles Role’s in Improving the Properties of the AA5250 Nanocomposites. Coatings2021, 11, 977. https://doi.org/10.3390/coatings11080977.
- On page 10, line 225, "Summary" should be "Conclusions".
Authors’ response
Manuscript has been corrected according to the comment
- The article contains numerous typographic and language errors. It should be corrected.
Authors’ response
According to the comment, the revised manuscript has been carefully spell-checked and corrected.
- The article should be rearranged by taking into account the journal writing rules and citation rules.
Authors’ response
According to the comment, the revised manuscript has been rearranged in accordance with the journal rules.
- Your reference list contains only one article from the "Materials" journal. If your work is appropriate for the context of this journal, there are many references from this journal. Second, the cited sources should be primary sources. So the indexed area shows the strength of a paper and directly the reliability of your paper. Please make arrangements in this direction.
Authors’ response
According to the comment, all papers on the subject that have been published in “Materials”, have been considered in the revised manuscript. The list of these articles is shown below.
- Gan, Y.X.; Solomon, D.; Reinbolt, M. Friction Stir Processing of Particle Reinforced Composite Materials. Materials2010, 3, 329-350. https://doi.org/10.3390/ma3010329.
- Huang, C.W.; Aoh, J.N. Friction Stir Processing of Copper-Coated SiC Particulate-Reinforced Aluminum Matrix Composite. Materials2018, 11, 599. https://doi.org/10.3390/ma11040599.
- Moustafa, E.B. Dynamic Characteristics Study for Surface Composite of AMMNCs Matrix Fabricated by Friction Stir Process. Materials2018, 11, 1240. https://doi.org/10.3390/ma11071240.
- Mahmoud E.R.I.; Tash M.M. Characterization of Aluminum-Based-Surface Matrix Composites with Iron and Iron Oxide Fabricated by Friction Stir Processing. Materials (Basel) 2016, 23, 505. doi: 10.3390/ma9070505.
- 30. Papantoniou, I.G.; Markopoulos, A.P.; Manolakos, D.E. A New Approach in Surface Modification and Surface Hardening of Aluminum Alloys Using Friction Stir Process: Cu-Reinforced AA5083. Materials2020, 13, 1278. https://doi.org/10.3390/ma13061278.
- Guo, L.; Liu, Y.; Shen, K.; Song, C.; Yang, M.; Kim, K.; Wang, W. Enhancing Corrosion and Wear Resistance of AA6061 by Friction Stir Processing with Fe78Si9B13Glass Particles. Materials 2015, 8, 5084-5097. https://doi.org/10.3390/ma8085084.
- Rubtsov, V.; Chumaevskii, A.; Gusarova, A.; Knyazhev, E.; Gurianov, D.; Zykova, A.; Kalashnikova, T.; Cheremnov, A.; Savchenko, N.; Vorontsov, A.; Utyaganova, V.; Kolubaev, E.; Tarasov, S. Macro- and Microstructure of In-Situ Composites Prepared by Friction Stir Processing of AA5056 Admixed with Copper Powders. Materials2023, 16, 1070. https://doi.org/10.3390/ma16031070.
- Chumaevskii, A.; Zykova, A.; Sudarikov, A.; Knyazhev, E.; Savchenko, N.; Gubanov, A.; Moskvichev, E.; Gurianov, D.; Nikolaeva, A.; Vorontsov, A.; Kolubaev, E.; Tarasov, S. In-Situ Al-Mg Alloy Base Composite Reinforced by Oxides and Intermetallic Compounds Resulted from Decomposition of ZrW2O8during Multipass Friction Stir Processing. Materials 2023, 16, 817. https://doi.org/10.3390/ma16020817.
Moreover, the preference in discussion of the literature sources has been given to original papers. Specifically, 49 out of 54 cited references were original works.
*** Authors must consider them properly before submitting the revised manuscript. A point-by-point reply is required when the revised files are submitted.
Authors’ response
Each Reviewer’s comment has been carefully considered by the authors. A detailed, point-by-point reply has been provided above.
Reviewer 2 Report
The manuscript mainly studies microstructural evoluation during the fabrication of Al/Al2O3 composite by friction stir processing (FSP), which is a meaningful study. However, the following questions and ambiguous descriptions should be carefully considered.
1. The abstract should be rewritten throughly. An absracte should state briefly the purpose of the research, the principal results and major conclusions.
2. The reference number in Introduction section (for example, [e.g., 2-18]) is incorrect, and the format in the Reference list also needs to be revised carefully.
3. Part of the experimental results is imcomplete, and the analysis of results is very concise and must be expanded. For example, what is the effect of Al2O3 on HAB fraction in Fig. 8b? Therefore, I would like to see the revised version of the paper and I hope that authors will revise carefully, doing a detailed analysis.
4. In my opinion, the microhardness maps in Fig. 4 lack readability. I suggest the data can be represented graphically in a line diagram, as shown in the following reference.
Fatigue and hardness effects of a thin buffer layer on the heat affected zone of a weld repaired Bisplate80. Journal of Materials Processing Technology, 2012, 212: 393.
Author Response
Responses to Reviewers’ comments on the paper entitled
“Microstructural aspects of the fabrication of Al/Al2O3 composite by friction stir processing” (Ms. Ref. No.: materials-2303515)
The authors would like to express their gratitude to Reviewer for useful remarks to improve the paper. Below, we provided specific replies to the issues raised.
The manuscript mainly studies microstructural evoluation during the fabrication of Al/Al2O3 composite by friction stir processing (FSP), which is a meaningful study. However, the following questions and ambiguous descriptions should be carefully considered.
- The abstract should be rewritten throughly. An absracte should state briefly the purpose of the research, the principal results and major conclusions.
Authors’ response
According to the comment, the abstract has been thoroughly revised. The recommended issues have been emphasized as follows (Page 1):
“Abstract: The purpose of this work was the examination of microstructural evolution during the fabrication of an Al/Al2O3 composite by friction stir processing (FSP). In order to get a new insight into this process, a longitudinal section of the produced composite was studied, and advanced characterization techniques (including electron backscatter diffraction and microhardness mapping) were applied. It was found that the reinforcing particles rapidly rearranged into the “onion-ring” structure, which was very stable against the subsequent dispersion. Specifically, the remnants of the comparatively coarse-scale particle agglomerations have survived even after 12 FSP passes. Therefore, it was concluded that three or four FSP passes, which are often applied in practice, are not sufficient to provide a homogeneous dispersion of the reinforcing particles. It was also revealed that the gradual distribution of the nanoscale Al2O3 particles throughout the aluminum matrix promoted a subtle reduction in both the portion of high-angle boundaries and the average grain size. These observations were attributed to the particle pinning of grain-boundary migration and dislocation slip.”
- The reference number in Introduction section (for example, [e.g., 2-18]) is incorrect, and the format in the Reference list also needs to be revised carefully.
Authors’ response
The manuscript has been carefully revised in accordance with the journal citation rules.
- Part of the experimental results is imcomplete, and the analysis of results is very concise and must be expanded. For example, what is the effect of Al2O3on HAB fraction in Fig. 8b? Therefore, I would like to see the revised version of the paper and I hope that authors will revise carefully, doing a detailed analysis.
Authors’ response
According to the comment, a more detailed discussion of experimental results has been provided in the revised manuscript. A particular emphasis has been given to the analysis of the influence of Al2O3 particles on the HAB fraction (Section 3.3.2, Paragraphs 2 to 4):
“After the first and second FSP passes, the important microstructural characteristic was the development of fine-grained bands (arrows in Fig. 8a and b). Remarkably, such bands (as well as the adjacent areas) often exhibited increased LAB content (Fig. 8b). These observations presumably reflected the influence of the nanoscale Al2O3 dispersoids.
Specifically, it is known that the relatively small (~1 mm in size) agglomerations of the nanoscale particles may promote the activation of the mechanism of the particle-stimulated recrystallization [6] and thus promote local grain refinement. On the other hand, the nano-scale dispersoids per se should retard grain-boundary migration and dislocation slip. In the first case, the enhancement of the grain-refinement effect is expected. The second effect may inhibit the progressive evolution of deformation-induced boundaries, including the LAB-to-HAB transformation; this should result in the decreased HAB fraction in the microstructure. It was therefore likely that the revealed fine-grained bands were associated with the local concentration of the Al2O3 particles.
Therefore, the gradual dispersion of the reinforcing particles throughout the aluminum matrix with sequential FSP passes resulted in a steady decrease in both grain size and HAB content (Figs. 8c-f and 9a-b). A gradual grain refinement within the matrix of the aluminum-based composites with the number of FSP passes has been reported in the scientific literature [16,17,20,21, 51-55]. “
References:
“6. Heidarzadeh, A.; Mironov, S.; Kaibyshev, R.; Cam, G.; Simar, A.; Gerlich, A.P.; Khodabakhshi, A.F.; Mostafaei, A.; Field, D.P.; Robson, J.D.; Deschamps, A.; Withers, P.J. Friction Stir Welding/Processing of Metals and Alloys: A Comprehensive Review on Microstructural Evolution. Prog. Mater. Sci. 2021, 117, 100752. doi:10.1016/j.pmatsci.2020.100752.
- Orlowska, M.; Pixner, F.; Hutter, A.; Enzinger, N.; Olejnik, L.; Lewandowska, M. Manufacturing of Coarse and Ultrafine-Grained Aluminum Matrix Composites Reinforced with Al2O3 Nanoparticles via Friction Stir Processing. J. Manuf. Proc. 2022, 80, 359-373. doi:10.1016/j.jmapro.2022.06011.
- Shafiei-Zarghani, A.; Kashani-Bozorg, S.F.; Zarei-Hanzaki, A. Wear Assessment of Al/Al2O3 Nanocomposite Surface Layer Produced Using Friction Stir Processing. Wear. 2011, 270, 403-412. doi://10.1016/j.weqr.2010.12.002.
- Moustafa, E. Effect of Multi-Pass Friction Stir Processing on Mechanical Properties for AA2024/Al2O3 Nanocomposites. Materials. 2017, 10, 1053. doi:10.3390/ma10091053.
- AbuShanab, W.S.; Moustafa, E.B. Effects of Friction Stir Processing Parameters on the Wear Resistance and Mechanuical Properties of Fabricated Metal Matrix Nanocomposites (MMNCs) Surface. J. Mater. Res. Technol. 2020, 9, 7460-7471. doi:10.1016/j.jmrt.2020.04.073.
- Moustafa, E.B.; AbuShanab, W.S.; Ghandourah, E.; Taha, M.A. Microstructural, Mechanical and Thermal Properties Evaluation of AA6061/Al2O3-BN Hybrid and Mono Nanocomposite Surface. J. Mater. Res. Technol. 2020, 9, 15486-15495. https://doi.org/10.1016/j.jmrt.2020.11.010.
- Moustafa, E.B.; Melaibari, A.; Alsoruji, G.; Khalil, A.M.; Mosleh, A.O. Tribological and Mechanical Characteristics of AA5083 Alloy Reinforced by Hybridising Heavy Ceramic Particles Ta2C & VC with Light GNP and Al2O3 Nanoparticles, Ceramic Inter. 2022, 48, 4710-4721. https://doi.org/10.1016/j.ceramint.2021.11.007.
- Moustafa, E.B.; Abushanab, W.S.; Melaibari, A.; Yakovtseva, O.; Mosleh, A.O. The Effectiveness of Incorporating Hybrid Reinforcement Nanoparticles in the Enhancement of the Tribological Behavior of Aluminum Metal Matrix Composites. JOM2021, 73, 4338–4348. https://doi.org/10.1007/s11837-021-04955-w.
- Abushanab, W.S.; Moustafa, E.B.; Melaibari, A.A.; Kotov, A.D.; Mosleh, A.O. A Novel Comparative Study Based on the Economic Feasibility of the Ceramic Nanoparticles Role’s in Improving the Properties of the AA5250 Nanocomposites. Coatings2021, 11, 977. https://doi.org/10.3390/coatings11080977.”
- In my opinion, the microhardness maps in Fig. 4 lack readability. I suggest the data can be represented graphically in a line diagram, as shown in the following reference.
Fatigue and hardness effects of a thin buffer layer on the heat affected zone of a weld repaired Bisplate80. Journal of Materials Processing Technology, 2012, 212: 393.
Authors’ response
The microhardness data has been rearranged in accordance with Reviewer’s recommendation. The evolution of the microhardness profile as a function of the number of FSP passes has been added to the revised manuscript as Figure 5. Appropriate discussion of the new data has been provided in Section 3.1 of the revised manuscript (Section 3.1):
“To assist interpretation of the macro-scale structures, microhardness data were also presented in Figs. 4 and 5. Specifically, the microhardness maps in Fig. 4 were used for examination of the spatial distribution of Al3O3 powder. By analogy with work by Zhang et al. [36], microhardness profiles were also measured to quantify the dispersion of the reinforcing particles through the matrix material (Fig. 5a). Moreover, the average microhardness of the acquired microhardness map was also shown as a function of the number of FSP passes in Fig. 5b.”
References
“36. Zhang, C.; Hu, X.; Lu, P. Fatigue and Hardness Effects of a Thin Buffer Layer on the Heat Affected Zone of a Weld Repaired Bisplate80, J. Mater. Proc. Technol. 2012, 212, 393-401. https://doi.org/10.1016/j.jmatprotec.2011.10.002. “

Round 2
Reviewer 1 Report
Journal: Materials (ISSN 1996-1944)
Manuscript ID: materials-2303515
Review Report 2#
The authors completed the requested corrections. In my opinion, this article is acceptable for final publication in the "Materials" journal.
Reviewer 2 Report
The suggested comments have been corrected in the revised manuscript.